# The Effect of Cognitive Task Complexity on Healthy Gait in the Walking Corsi Test

**DOI:** 10.3390/brainsci13071019

**Published:** 2023-06-30

**Authors:** Nicola Camp, Roberto Vagnetti, Maria Bisele, Paul Felton, Kirsty Hunter, Daniele Magistro

**Affiliations:** School of Science and Technology, Nottingham Trent University, Nottingham NG11 8NS, UK; roberto.vagnetti@ntu.ac.uk (R.V.); maria.bisele@ntu.ac.uk (M.B.); paul.felton@ntu.ac.uk (P.F.); kirsty.hunter@ntu.ac.uk (K.H.); daniele.magistro@ntu.ac.uk (D.M.)

**Keywords:** ADLs, visual–spatial memory, navigation, IMU sensors

## Abstract

Dual-task activities are essential within everyday life, requiring visual–spatial memory (VSM) and mobility skills. Navigational memory is an important component of VSM needed to carry out everyday activities, but this is often not included in traditional tests such as the Corsi block tapping test (CBT). The Walking Corsi Test (WalCT) allows both VSM and navigational memory to be tested together, as well as allowing measures of gait to be collected, thus providing a more complete understanding of dual-task function. The aim of this study was to investigate the effect of an increasingly complex cognitive task on gait in a healthy adult population, using the WalCT and body-worn inertial measurement unit (IMU) sensors. Participants completed both the CBT and WalCT, where they were asked to replicate increasingly complex sequences until they were no longer able to carry this out correctly. IMU sensors were worn on the shins throughout the WalCT to assess changes in gait as task complexity increased. Results showed that there were significant differences in several gait parameters between completing a relatively simple cognitive task and completing a complex task. The type of memory used also appeared to have an impact on some gait variables. This indicates that even within a healthy population, gait is affected by cognitive task complexity, which may limit function in everyday dual-task activities.

## 1. Introduction

People require visual spatial memory skills to recall and process an object’s identity and spatial location as they move [1]. These skills are essential to perform activities of daily living (ADLs) and thus live independently [2,3]. Visual spatial memory (VSM) skills could be conceptualised into short-term (VSSM) and working memory components (VSWM) [4]. The short term allows for short retention and the immediate recall of an object’s identity and spatial location, whereas the working component allows the use of the retained information for the execution of cognitive tasks, such as problem solving [4,5,6]. VSWM comprises three elements [7]: (1) the central executive, which controls and regulates cognitive processes such as troubleshooting and planning, (2) a phonological loop, which stores verbal or acoustic information, and (3) a visual–spatial sketchpad, which stores visual or spatial information [8]. These three elements allow people to understand where they are within an environment, and where objects are in relation to themselves [9]. Although there is evidence that VSWM declines with age [9,10,11], which may impact an individual’s ability to live independently, there is little known about how an individual’s movements are impacted as VSWM complexity increases.

The Corsi block tapping test (CBT) is often considered the gold-standard method of assessing VSSM and VSWM [4]. During the CBT, a pattern of nine blocks is placed in front of the participant, and they are asked to replicate a series of sequences initially tapped out on the blocks by a tester. Two variations are often used to assess VSSM and VSWM: (1) the forward Corsi (testing VSSM), which requires the sequence to be replicated in the same order it was presented; (2), the backward Corsi (testing VSWM), which requires the sequence to be replicated in the reverse of the order it was presented in [11]. In both variations, the number of blocks increases by one for each sequence, increasing the complexity of the test. This increase allows for the participant’s limit to be established as they will reach a point where they can no longer replicate the sequence correctly., There are many difficulties, however, when assessing VSSM and VSWM, due to the variation in the definition and theory used to explain how each component functions [12]. A limitation of the CBT is that it primarily assesses memory components in the area in front of one’s body where objects can be physically touched also known as the frontal peri-personal space [13]. It does not allow visual spatial memory skills to be assessed when navigation is required [9]. The consideration of the effect of navigation is important when investigating ADLs as individuals are often required to navigate through the home during these activities. For example, there is a need to navigate to the kitchen to prepare food.

To assess visual spatial memory skills during activities requiring navigation, the Walking Corsi Test (WalCT) was developed [9,14,15]. The WalCT follows a similar protocol to that of the CBT but in a vista space setting where a wider space can be viewed from a single location [16]. Instead of replicating the sequence by tapping blocks as in the CBT, the sequence is demonstrated and replicated using a pattern of squares placed on the floor and walking between them. Previous research has suggested that the WalCT has also been able to detect deficits in navigational memory, even when no deficits in other aspects of VSM are present [9,11,17,18]. When used together, the CBT and WalCT allow visuo–spatial memory to be assessed and compared within both a peri-personal and vista space setting [9,11,14,15,18,19,20,21,22]. This provides a more comprehensive understanding of visual spatial memory function to be determined compared to that using the CBT in isolation.

Previous research has investigated how visual spatial memory affects individual movement patterns. It is well-documented that gait patterns are negatively affected by the addition of a cognitive task being performed at the same time especially in older adults [8,23,24,25,26,27,28,29]. Stride length and gait speed were found to be reduced in these dual-task activities (movement + cognitive element) versus a single-task activity (movement only), while stride time and gait variability were observed to increase [25,26,27,29]. A potential explanation for these differences may be that postural control is heavily influenced by visual input. Visual input uses the same information processing pathway as one of the VSWM elements required for navigation—the visuo-spatial sketchpad [8]. This may result in there being insufficient cognitive resources available to cope with the increased cognitive load of having two concurrent tasks operating within the same pathway [24]. There has been little consistency, however, in the cognitive activity utilised. For example, Grabiner and Troy [30] used the Stroop task, to assess both the central executive and visuo-spatial sketchpad elements of VSWM but provided no indication of which affected performance. Although there were no reported differences in step width with the addition of a cognitive task, more errors occurred in the Stroop task during the dual-task condition compared to the control. In contrast, Qu [8] used three separate tests, to investigate each separate element of VSWM (random number generation using the central executive; Brooks spatial memory using the visuo-spatial sketchpad; and counting Backward using the phonological loop). This test showed that gait measures such as step time and step width variability increased during the dual task compared to those during the control condition, which was linked to a need to control postural stability by reducing anterior velocity [8]. It should also be noted that both studies used a treadmill instead of overground walking, which is also known to affect gait patterns [31]. These studies therefore may not demonstrate the true interaction between visual spatial memory and gait in a real-world environment.

To investigate the effect of cognitive task complexity on gait patterns, a more appropriate approach would be to use a dual-task test which incorporates overground walking and participants with ‘healthy’ gait, such as the WalCT. Although the WalCT is a visual spatial memory test which utilises overground walking, it has not been utilised to investigate how visual spatial memory skills impact gait patterns. This is possible since the complexity of the task changes with the length of the sequence. As the sequences increase, the test is perceived to become more difficult as the cognitive resources required to maintain both walking and navigation tasks increase [32]. Although the perceived difficulty will differ between individuals based on their underlying cognitive function and ability, the WalCT allows the task to gradually increase in cognitive complexity to observe the level of failure for each individual and how this affects gait movement patterns. Initially, the WalCT can be considered to have a low cognitive load, as the sequences to be replicated are very short. As the length of the presented sequences increases, so does the cognitive load. The purpose of this study, therefore, is to investigate the effect of cognitive task complexity on gait patterns using a test which incorporates overground walking (WalCT) within a healthy population. It is hypothesised that as the cognitive load increases, gait movement patterns will alter as attention shifts towards completing the cognitive task.

## 2. Materials and Methods

### 2.1. Study Design

This study followed a cross-sectional, within-groups design. Data was collected in one of the biomechanics laboratories at Nottingham Trent University between June 2021 and May 2022. Power analysis using G-Power software showed that 36 participants were required to gather 80% power with α = 0.05. To account for attrition, we aimed to recruit ~40 participants for this study.

### 2.2. Participants

A total of 43 healthy adults, 20 males (age: 30.4 ± 9.2 years; height: 1.80 ± 0.7 m; weight: 82.3 ± 11.9 kg) and 23 females (age: 30.0 ± 8.8 years; height: 1.67 ± 0.7 m; weight: 64.8 ± 9.2 kg), participated in this study. All participants were recruited via convenience sampling and had self-reported good health status, defined as the absence of any injury, illness, or lower limb pathologies which would affect their participation in the study. The testing procedures were explained to each participant prior to participation in accordance with Nottingham Trent University’s ethical guidelines and informed consent was obtained.

### 2.3. Data Collection

Participants were required to complete both the Corsi block tapping test and walking Corsi test (Figure 1).

Prior to completing the WalCT, each participant completed a CBT conducted by an experienced practitioner (NC). The participant and practitioner sat at a table opposite each other with a CBT board consisting of nine blocks (4.5 × 9 × 4.5 cm) fixed on a baseboard (30 × 25 cm) arranged in a standardised scattered array (Figure 1a) placed between them [9,14]. Each test was initiated by the practitioner who demonstrated a pre-defined sequence of blocks (Appendix A) by tapping each block with one finger at a rate of one block per 2 seconds The participant was then asked to repeat the sequence, using one finger to tap each block in turn, firstly in the sequence demonstrated (forward condition) and secondly in reverse (backward condition). Two familiarisation sequences were conducted at the beginning of both the forward and backward conditions to ensure participants understood. To increase the cognitive complexity of the CBT, the number of blocks in the sequence gradually increased. Each participant was given a maximum of three sequences at each given length, with participants progressing to the next sequence length if two sequences of the same length were successfully replicated. If the participant was unable to replicate two sequences of the same length, the CBT condition was ended. For each condition, the success of each attempted sequence and block span (the maximum length of the sequence at which the participant was able to replicate two successfully) were recorded.

The WalCT was conducted by the same experienced practitioner (NC), and testing was completed in a secluded environment to prevent participants using any external landmarks to aid the completion of the cognitive task. The participant began each test sat in a chair with both feet flat on the floor and knees flexed at approximately 90°, The WalCT board, consisting of a nine large squares (45 × 45 cm) fixed on a large mat (4.5 × 3.75 m; scale, 1:15 to the CBT) and arranged in the same standardised CBT scatter array, was located in front of them (Figure 1b). Each test was initiated by the practitioner who demonstrated a pre-defined sequence by walking between squares and pausing on each included square for 2 seconds before returning to the initial location (stood in front of the participant’s chair) and exiting the mat. The participant was then asked to repeat the sequence, by walking between the squares and pausing for approximately 1–2 seconds on the square which they thought was next in the sequence. Once participants felt they had replicated the sequence, they were asked to return to their original seated position. Participants were allowed to walk anywhere on the mats (including crossing over squares) but were instructed to face the direction they were travelling in and to not jump between the squares. Similarly to the CBT, participants completed sequence recollection in both the forward and backward conditions (Figure 2) and were provided with two familiarisation sequences at the start of each condition to ensure they understood. The cognitive complexity of the test was also increased using the same method of gradually increasing the number of blocks in the sequence, with the WalCT condition ending if participants could not replicate two sequences out of three successfully at each length. For each condition, the success of each attempted sequence and block span (the maximum length of the sequence at which the participant was able to replicate two successfully) were recorded. In addition, kinematic gait data were captured during the WalCT (between the practitioner leaving the mat and the participant returning to a seated position) using two Kinesis IMU sensors (Kinesis Health Technologies, Dublin, Ireland) attached anteriorly to the centre of the participants’ tibias using a Velcro strap.

### 2.4. Data Processing

Nine spatial and temporal parameters relating to gait were computed using the Kinesis software [33] for the first three successful sequences in each condition and the last failed trial to investigate the effect of cognitive task complexity on gait (e.g., if a participant reached a block span of 6, the trials for analysis were the first two successful trials at level 1, the first successful trial at level 2, and the last unsuccessful trial at level 7). This approach allowed the differences in gait to be compared between conditions with a low cognitive load initially (the initial successful trials) to initially identify whether or not there were significant differences in natural gait parameters independent of the task itself. The subsequent comparison between conditions with a low cognitive load (initial successful trials) and high cognitive load (last failed level) allowed the effect of cognitive load on healthy gait to be determined. Nine parameters were determined for each test within the Kinesis software (QTUGTM results interpretation and guidance, version 4.1):*Average double support* (%)—proportion of the gait cycle spent on both feet;*Average single support* (%)—proportion of the gait cycle spent on either foot;*Average stance time* (s)—average time between heel strike and toe-off on each foot;*Average step time* (s)—average time between heel strike on one foot, and heel strike on the opposite foot;*Average stride time* (s)—average time between successive heel strikes on the same foot;*Average stride velocity* (cm/s)—average walking speed;*Average swing time* (s)—average time between toe-off and heel strike of the same foot;*Cadence* (steps/min)—average number of steps taken per minute;*Time to stand* (s)—time taken from initiating movement to first heel strike or toe-off on either foot.

### 2.5. Statistical Analysis

Statistical analysis was performed using the SPSS computer package (IBM, Chicago, IL, USA) using an alpha threshold value of 0.05 to determine statistical significance. A preliminary ANCOVA analysis was conducted to explore the effect of the individual test level at which participants were no longer able to replicate two sequences of the same length correctly (failure level) on the gait parameters, and the effect of gender on block span and gait parameters. No significant effects were observed between the failure level and the gait parameters; however, gender was found to be significantly related to step time during the forward condition. Repeated measures ANCOVA analysis was then used to compare CBT and WalCT results, in both the forward and backward conditions. Separate repeated measures ANCOVA, with Greenhouse–Geisser correction where a lack of sphericity occurred, were used to explore differences in the selected gait parameters between the initial three successful trials (S1, S2 and S3) and the final failed trial (fail) in both the forward and backward conditions. Significant interactions within each ANCOVA were identified using Bonferroni post hoc comparisons, and effect sizes were calculated using Cohen’s d, where 0.2 = small effect, 0.5 = medium effect and 0.8 = large effect.

## 3. Results

### 3.1. Block Span

Repeat measures ANCOVA analysis (Table 1) showed significant differences between the different trials (F(3,112) = 5.45, η^2^ = 0.117, *p* < 0.01). Bonferroni post hoc pairwise comparisons showed that there were significant differences between the forward CBT and forward WalCT (*p* < 0.001, d = 2), forward CBT and backward WalCT (*p* < 0.001, d = 2), forward WalCT and backward CBT (*p* < 0.001, d = 1) and between the backward CBT and backward WalCT (*p* < 0.001, d = 1). There were no significant differences between the forward CBT and backward CBT or between the forward WalCT and backward WalCT. Gender was also shown to have no significant interaction. These results indicate that the WalCT is more difficult for the participants than the CBT task.

### 3.2. Gait Parameters

Repeat measures ANCOVA analyses were used to identify significant differences between trials with a low cognitive load (the first three successful trials; “S1”, “S2”, “and S3”) and a trial with a high cognitive load (the final unsuccessful trial; “fail”) of both the forward WalCT (Table 2) and backward WalCT (Table 3). There were significant differences in all variables within the forward condition except step time (Figure 3), and in all variables in the backward condition except single support, stance time and swing time (Figure 4).

#### 3.2.1. Forward Condition

There were significant differences between the high cognitive load trial and all three trials with a low cognitive load in double support ((F(3,118) = 8.50, η^2^ = 0.168, *p* < 0.001); fail vs. S1, *p* < 0.001, d = 0.7; fail vs. S2, *p* = 0.047, d = 0.5; fail vs. S3, *p* = 0.002, d = 0.6), and stride velocity ((F(3,117) = 13.38, η^2^ = 0.242, *p* < 0.001]; fail vs. S1, *p* = 0.006, d = 0.6; fail vs. S2, *p* < 0.001, d = 1.0; fail vs. S3, *p* < 0.001, d = 0.7), and time to stand ((F(2,72) = 14.53, η^2^ = 0.257, *p* < 0.001); fail vs. S1, *p* = 0.003, d = 0.9; fail vs. S2, *p* < 0.001, d = 1.0; fail vs. S3, *p* = 0.001, d = 0.9).

There were significant differences between the high cognitive load trial and one of the low cognitive load trials (S3), and between two of the low cognitive load trials (S1 verse S3) in step time (F(3,106) = 6.52, η^2^ = 0.134, *p* < 0.001); fail vs. S3, *p* = 0.012, d = 0.6; S1 vs. S3, *p* = 0.001, d = 0.7), swing time ((F(2,100) = 6.57, η^2^ = 0.135, *p* = 0.002); fail vs. S3, *p* = 0.013, d = 0.5; S1 vs. S3, *p* < 0.001, d = 0.7) and stance time ((F(2,100) = 6.57, η^2^ = 0.135, *p* = 0.002); fail vs. S3, *p* = 0.013, d = 0.5; S1 vs. S3, *p* = 0.001, d = 0.7).

Cadence showed significant differences ((F(3,121) = 31.96, η^2^ = 0.432, *p* < 0.001_) between multiple trials: S1 vs. S2 (*p* = 0.009, d = 0.5), S1 vs. S3 (*p* < 0.001, d = 1.1), S2 vs. S3 (*p* = 0.001, d = 0.6), S2 vs. fail (*p* < 0.001, d = 0.8) and S3 vs. fail (*p* < 0.001, d = 1.3). Stride time also showed significant differences (F(3,118) = 15.51, η^2^= 0.270, *p* < 0.001) across multiple trials: S1 vs. S2 (*p* = 0.013, d = 0.4), S1 vs. S3 (*p* < 0.001, d = 0.9), S2 vs. S3 (*p* = 0.002, d = 0.5) and S3 vs. fail (*p* < 0.001, d = 0.7).

Single support showed no significant differences between any trials within the forward condition. When considering gender as a covariate, step time was the only variable to show some significant interaction (F(3,113) = 6.27, η^2^ = 0.133, *p* < 0.001), with men demonstrating a significant difference between S1 and S2 (*p* < 0.05 d = 1.0), S1 and S3 (*p* < 0.01, d = 1.2), and S1 and fail (*p* < 0.05, d = 0.9) whereas women only showed a difference between fail and S3 (*p* < 0.01, d = 0.5).

#### 3.2.2. Backward Condition

There were significant differences between the high cognitive load trial and all three low cognitive load trials (fail vs. S1, S2 and S3) in double support ((F(3,103) = 22.04, η^2^ = 0.361, *p* < 0.001); fail vs. S1, *p* < 0.001, d = 1.3; fail vs. S2, *p* < 0.001, d = 1.4; fail vs. S3, *p* < 0.001, d = 1.0), stride time ((F(2,86) = 17.80, η^2^ = 0.313, *p* < 0.001); fail vs. S1, *p* < 0.001, d = 1.1; fail vs. S2, *p* = 0.001, d = 0.8; fail vs. S3, *p* < 0.001, d = 0.9) and time to stand ((F(1,47) = 18.54, η^2^ = 0.322, *p* < 0.001); fail vs. S1, *p* = 0.001, d = 0.9; fail vs. S2, *p* < 0.001, d = 0.9, fail vs. S3, *p* < 0.001, d = 0.9).

Cadence showed significant differences (F(2,71) = 34.78, η^2^ = 0.471, *p* < 0.001) between multiple trials: S1 vs. S2 (*p* = 0.014, d = 0.4), S1 vs. fail (*p* < 0.001, d = 1.6), S2 vs. fail (*p* < 0.001, d = 1.3), and S3 vs. fail (*p* < 0.001, d = 1.4). Stride velocity also showed significant differences [F(3,104) = 33.82, η^2^ = 0.464, *p* < 0.001] between multiple trials: S1 vs. S2 (*p* < 0.001, d = 1.3), S1 vs. S3 (*p* = 0.011, d = 1.3), S2 vs. S3 (*p* < 0.001, d = 0.8), S2 vs. fail (*p* < 0.001, d = 1.8) and S3 vs. fail (*p* < 0.001, d = 1.0).

Step time showed significant differences (F(3,102) = 6.28, η^2^ = 0.139, *p* < 0.001] between the high-cognitive-load trial, the first low-cognitive-load trial (S1: *p* = 0.013, d = 0.7) and third low-cognitive-load trial (S3: *p* = 0.007, d = 0.7). There were no significant differences between trials during the backward task in single support, stance time or swing time.

## 4. Discussion

This study aimed to investigate how increasing the cognitive load within the WalCT influences gait within a healthy adult population. In line with the hypothesis, there were significant differences in several gait parameters when comparing trials with low and high cognitive loads in both the forward and backward WalCT conditions. The specific failure level reached by each participant did not have any influence on the results, suggesting that gait is not influenced by an individual’s proficiency but by the increasing perceived difficulty of the task itself. There was more variation in the gait variables within the forward condition, with significant difference present between at least two trials in all gait parameters-apart from the single support time which showed no significant differences between any trials. In contrast, the backward condition showed no significant differences in single support, swing time or stance time. There were also fewer differences between the trials with a low cognitive load in the remaining variables. The effect sizes between the failed trial and initial successful trials were also larger in the backward trials, with most demonstrating a large effect size (d > 0.8). This suggests that working navigational memory may have a larger impact on gait variables than short term navigational memory does. In line with previous studies [9,14], significant differences in block span between the CBT and WalCT in both the forward and backward conditions were observed, with participants performing better in the CBT compared to the WalCT. This adds to the theory that even if the two tasks have some common components, each test measures different aspects of memory [9], with the WalCT generally presenting a larger span than CBT does [14,15]. This may be linked to differences in the underlying processes related to peri- and extra-personal space [15], and individual navigational deficits which appear in the WalCT but are either not present or less prevalent in the CBT [17].

No significant interaction between the specific fail level and the gait parameters in either the forward or backward WalCT condition were observed. This suggests that the components of VSWM required for gait and navigation are linked regardless of an individual’s memory capability. Those with a longer block span, and therefore those considered to have greater memory capability still showed significant differences in gait parameters. This adds to the existing theory that gait and cognitive processes are closely intertwined [34]. Many existing studies focusing on the relationship between VSWM and gait, rarely include measures outside of gait speed [34]. The utilisation of IMU sensors allowing a more detailed analysis of differences in gait parameters to be determined throughout a complex task may help to explain this connection. Changes in gait are often minimal and may not be detected by even a well-trained eye, instead requiring the use of a computerised tool [34]. To date, marker-based motion capture systems have been used to assess gait changes during dual-task walking performance [8]; however, these systems are complex and time-consuming to use. The addition of inertial measurement unit (IMU) sensors, which typically comprise different microsensors such as an accelerometer and gyroscope, attached to the shin have been shown to detect these minimal changes without the physical constraints and high costs typically associated with laboratory-based equipment such as video-based motion capture and force plates [33,35]. There is potential for the WalCT to be conducted outside of a laboratory-based setting, and therefore IMU sensors have the potential to provide valuable data where other systems may not be suitable; however, this would need to be investigated in future work.

Average double support and time to stand were significantly different during both the forward and backward WalCT tasks. In both instances, they were significantly longer in the failed task compared to all three successful trials. Interestingly, there was no significant difference between any of the successful trials. The increase in double support time suggests that as the cognitive task becomes more difficult there is a need to shift cognitive resources from a primary task (walking) to the secondary task (route recollection), and therefore locomotion is temporarily stopped. The increase in time taken to stand may also imply that participants were attempting to recall as much of the shown route as possible before adding in the additional task of controlling locomotion. Ellmers et al. [23] suggest that this may be due to a shift in attention from external information related to locomotion and the environment to internal information relating to movement control. This is especially prevalent in older adults who tend to “stop walking when talking”, with these individuals showing a greater inclination to consciously control their movements while simultaneously showing a poorer retention of visuo-spatial information. The fact that a younger cohort also displayed this tendency to increase their double support time suggests that it is a useful strategy to aid in the completion of a difficult cognitive task, regardless of age-related factors. It would be interesting, however, to compare the results of the younger group used in this study to those of an older cohort to investigate whether or not this strategy occurs sooner in older adults.

When comparing gait parameters, it can be difficult to differentiate natural variation in gait from those variations caused by the task. In this study (Table 2 and Table 3), there were significant differences in most parameters between at least two trials. The key differences, however, were mostly between the trial with a high cognitive load and trials with a low cognitive load. Previous studies have suggested that as a cognitive task within a dual-task activity is perceived as more complex, the attention shifts away from walking, as it is considered an autonomous activity [24]. In a younger population, such as the one used in this study, this shift is perhaps not as difficult to cope with; however, cognitive ability is known to decline with age [36]. Goal-oriented locomotion, such as that required to complete most ADLs, requires cognitively demanding processes [37]. Alongside the cognitive decline adults may experience as they age [36], age-related cognitive impairments could influence individuals’ motor skills [38,39]. A study by Oliveira et al. [40] indicated that older adults show reduced motor performance during a complex cognitive task compared to that of young adults. Ageing is also linked to a decline in walking ability, and an increase in issues such as a fear of falling may mean that more attention is needed to control gait in everyday life causing a worsening of their locomotion [41], regardless of the presence of an additional cognitive task. The population used within this study aimed to understand some of these differences in gait as cognitive complexity increased without the possible influence of age-related decline in either physical or cognitive ability; this should be a focus of future work.

During ADL assessments, the physical and cognitive elements of an activity are often tested separately [2], with the dual-task nature of the activities in question being largely ignored. To carry out activities such as cooking for example, an individual needs to be able to complete a series of complex activities including controlling their body position, manoeuvring around a kitchen, and keeping track of the sequence of events needed to create a meal. This study highlights that, even in a younger cohort, movements slow down as cognitive load increases. In older adults, this increase in cognitive load has been linked to reduced performance during dual-task activities [40] and an increased fall risk in individuals with cognitive impairment [42]. Deficits in executive functioning, which include VSWM, have also been linked to the early stages of Alzheimer’s disease [43]. Future studies should focus on understanding how the WalCT combined with wearable sensors to assess gait can be used with an older cohort, to highlight how potential age-related issues may influence performance.

There are a few notable limitations within this study. Firstly, the WalCT is still a lab-based study and therefore may not relate fully to an individual’s living situation. Additional environmental cues which exist in a standard living environment may assist performance; however, this has yet to be studied. Secondly, the population used within this study comprised healthy adults who typically would not be expected to display any issues with gait or the types of memory being tested. To fully understand this connection, a wider range of participants should be used; e.g., one including older adults, or those with physical/cognitive impairments.

## 5. Conclusions

In conclusion, this study has shown that there are significant differences in several gait variables when the cognitive aspect of a dual-task activity becomes too complex. There also appears to be differences in the type of memory being used to complete a given task—whether the short-term or working memory component was utilised led to differences in some gait variables. Overall, movements were slowed when cognitive load increased, even within a young and healthy population. It is possible that these changes in movement will be greater in an older adult population, which may have further consequences for activities of daily living, and subsequently the ability to live independently.

## Figures and Tables

**Figure 1 brainsci-13-01019-f001:**
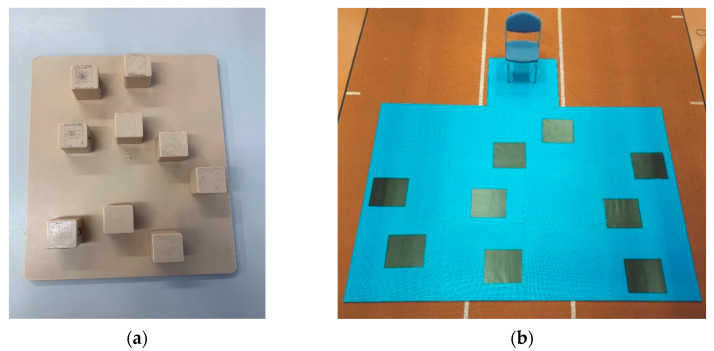
(**a**) The experimental setup and configuration of the CBT; (**b**) the experimental setup and configuration of the WalCT.

**Figure 2 brainsci-13-01019-f002:**
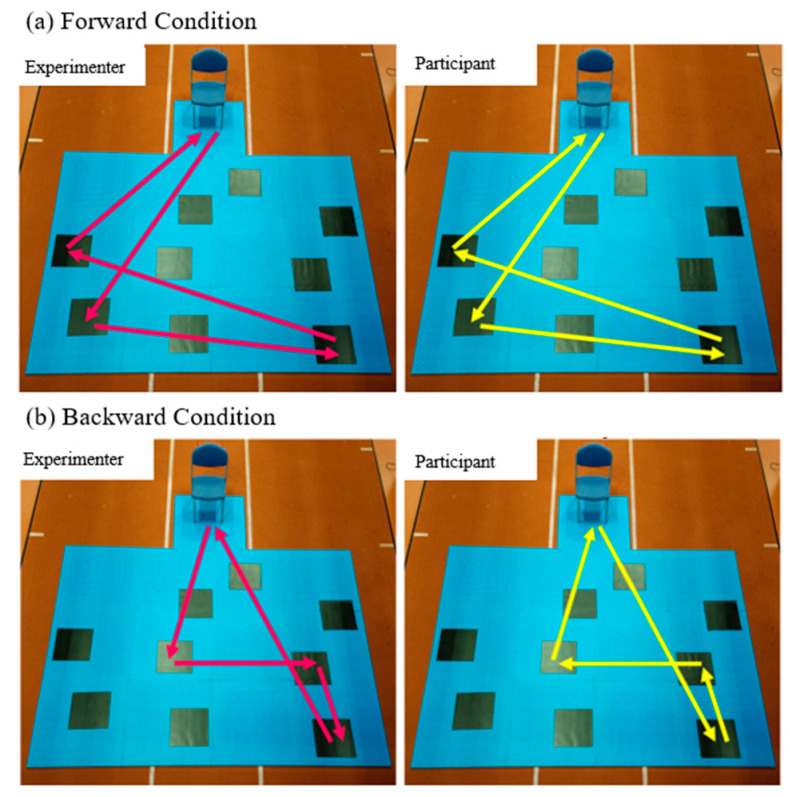
Examples of the forward and backward conditions for one sequence in the WalCT. Pink lines (**left**) are the sequence performed by the experimenter; yellow lines (**right**) represent the sequence performed by the participant.

**Figure 3 brainsci-13-01019-f003:**
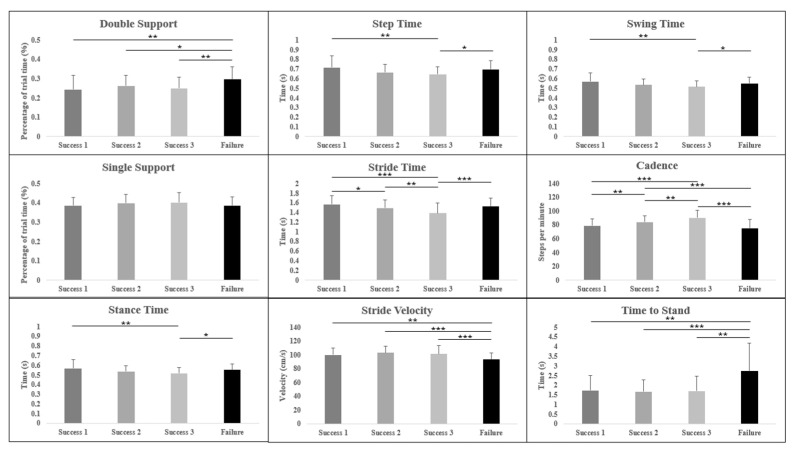
Summary of the gait parameters measured during the forward WalCT task, showing the differences between the first three successful trials and last failed trial. * *p* < 0.05, ** *p* < 0.01, and *** *p* < 0.001.

**Figure 4 brainsci-13-01019-f004:**
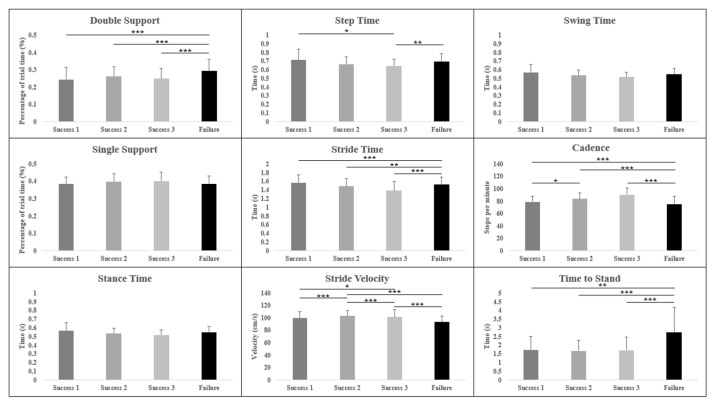
Summary of the gait parameters measured during the backward WalCT task, showing the differences between the first three successful trials and last failed trial. * *p* < 0.05, ** *p* < 0.01, and *** *p* < 0.001.

**Table 1 brainsci-13-01019-t001:** Block span ANOVA results, including Bonferroni post hoc pairwise comparisons. * *p* < 0.05.

Trial Condition (M; SD) (Median; IQR)				ANCOVA	Pairwise Comparison
Forward CBT	Forward WalCT	Backward CBT	Backward WalCT		
6;1 (6;1)	4;1 (4;1)	5;1 (5;2)	4;1 (4;1)	*F* = 5.45; *p* < 0.01; η^2^ = 0.117	ForwCBT > ForwWalCT * ForwCBT > BackWalCT * BackCBT > ForwWalCT * BackCBT > BackWalCT *

**Table 2 brainsci-13-01019-t002:** Forward condition WalCT gait variable ANOVA results, including Bonferroni post hoc pairwise comparisons. * *p* < 0.05.

Gait Variable	Trial Level (M; SD)				ANCOVA	Pairwise Comparison (*p*-Value) d
	Successful 1 (S1)	Successful 2 (S2)	Successful 3 (S3)	Unsuccessful (Fail)		
Forward Trials						
Average Double Support (%)	0.24; 0.07	0.26; 0.05	0.25; 0.06	0.29; 0.07	*F* = 8.50 *p* < 0.001 * η^2^ = 0.168	S1-fail * (0.001) 0.7 S2-fail * (0.047) 0.5 S3-fail * (0.002) 0.6
Average Single Support (%)	0.39; 0.04	0.40; 0.05	0.40; 0.05	0.39; 0.57	*F* = 1.96 *p* > 0.05 η^2^ = 0.045	-
Average Stance Time (s)	0.57; 0.09	0.54; 0.06	0.52; 0.06	0.55; 0.06	*F* = 6.57 *p* = 0.002 * η^2^ = 0.135	S1-S3 * (0.001) 0.7 S3-fail * (0.013) 0.5
Average Step Time (s)	0.72; 0.12	0.66; 0.09	0.65; 0.08	0.70; 0.09	*F* = 6.52 *p* < 0.001 * η^2^ = 0.134	S1-S3 * (0.001) 0.7 S3-fail * (0.012) 0.6
Average Stride Time (s)	1.57; 0.19	1.49; 0.17	1.39; 0.21	1.53; 0.18	*F* = 15.51 *p* < 0.001 * η^2^ = 0.270	S1-S2 * (0.013) 0.4 S1-S3 * (<0.001) 0.9 S2-S3 * (0.002) 0.5 S3-fail * (<0.001) 0.7
Average Stride Velocity (cm/s)	99.80; 10.09	103.20; 9.01	101.32; 12.57	93.41; 9.82	*F* = 13.38 *p* < 0.001 * η^2^ = 0.242	S1-fail * (0.006) 0.6 S2-fail * (<0.001) 1.0 S3-fail * (<0.001) 0.7
Average Swing Time (s)	0.57; 0.09	0.54; 0.06	0.52; 0.06	0.55; 0.06	*F* = 6.57 *p* < 0.002 * η^2^ = 0.135	S1-S3 * (0.001) 0.7 S3-fail* (0.013) 0.5
Cadence (steps/min)	78.96; 9.45	84.01; 9.48	90.40; 11.25	75.24; 12.48	*F* = 31.96 *p* < 0.001 * η^2^ = 0.432	S1-S2 * (0.009) 0.5 S1-S3 * (<0.001) 1.1 S2-S3 * (0.001) 0.6 S2-fail * (<0.001) 0.8 S3-fail * (<0.001) 1.3
Time Taken to Stand (s)	1.74; 0.77	1.67; 0.62	1.70; 0.77	2.73; 1.45	*F* = 14.53 *p* < 0.001 * η^2^ = 0.257	S1-fail * (0.003) 0.9 S2-fail * (<0.001) 1.0 S3-fail * (0.001) 0.9

**Table 3 brainsci-13-01019-t003:** Backward condition WalCT gait variable ANOVA results, including Bonferroni post hoc pairwise comparisons. * *p* < 0.05.

Gait Variable	Trial Level (M; SD)				ANCOVA	Pairwise Comparison (*p*-Value) d
	Successful 1 (S1)	Successful 2 (S2)	Successful 3 (S3)	Unsuccessful (Fail)		
Backward Trials						
Average Double Support (%)	0.22; 0.08	0.23; 0.05	0.24; 0.08	0.31; 0.06	*F* = 22.04 *p* < 0.001 * η^2^ = 0.361	S1-fail * (<0.001) 1.3 S2-fail * (<0.001) 1.4 S3-fail * (<0.001) 1.0
Average Single Support (%)	0.41; 0.05	0.40; 0.05	0.41; 0.06	0.39; 0.04	*F* = 1.98 *p* > 0.05 η^2^ = 0.048	-
Average Stance Time (s)	0.54; 0.08	0.55; 0.07	0.53; 0.07	0.56; 0.06	*F* = 6.57 *p* > 0.05 η^2^ = 0.135	-
Average Step Time (s)	0.65; 0.09	0.67; 0.09	0.65; 0.08	0.71; 0.09	*F* = 6.28 *p* < 0.001 * η^2^ = 0.139	S1-fail * (0.013) 0.7 S3-fail * (0.007) 0.7
Average Stride Time (s)	1.37; 0.19	1.42; 0.18	1.37; 0.24	1.57; 0.18	*F* = 17.80 *p* < 0.001 * η^2^ = 0.313	S1-fail * (<0.001) 1.1 S2-fail * (0.001) 0.8 S3-fail * (<0.001) 0.9
Average Stride Velocity (cm/s)	93.82; 13.34	108.99; 10.20	100.43; 11.00	89.57; 11.51	*F* = 33.82 *p* < 0.001 * η^2^ = 0.464	S1-S2 * (<0.001) 1.3 S1-S3 * (0.011) 1.3 S2 -S3 (<0.001) 0.8 S2- fail * (<0.001)1.8 S3-fail * (<0.001) 1.0
Average Swing Time (s)	0.54; 0.08	0.55; 0.07	0.53; 0.07	0.56; 0.06	*F* = 1.90 *p* > 0.05 η^2^ = 0.047	-
Cadence (steps/min)	93.34; 11.35	88.92; 10.09	91.85; 13.34	73.47; 13.77	*F* = 34.78 *p* < 0.001 * η^2^ = 0.471	S1-S2 * (0.014) 0.4 S1-fail * (<0.001) 1.6 S2-fail * (<0.001) 1.3 S3-fail * (<0.001) 1.4
Time Taken to Stand (s)	1.42; 0.48	1.33; 0.54	1.34; 0.58	2.85; 2.27	*F* = 18.54 *p* < 0.001 * η^2^ = 0.322	S1-fail * (0.001) 0.9 S2-fail * (<0.001) 0.9 S3-fail * (<0.001) 0.9

## Data Availability

The data presented in this study are available upon request from the corresponding author.

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
