# Peer review of "The Effect of Cognitive Task Complexity on Healthy Gait in the Walking Corsi Test"

_brainsci, 2023, doi:10.3390/brainsci13071019_

Round 1

Reviewer 1 Report

I had the opportunity to review this interesting study evaluating the effects of the complexity of cognitive tasks on gait in a sample of healthy adults. The manuscript is well-written, the study is novel and relevant, and it is methodologically sound. I have only four minor comments to improve the manuscript’s quality. 

a. Please add 1-2 sentences about the study’s methods in the abstract. 

b. I would suggest adding a “design” subheading before the “participants” subheading in the methods section and adding the relevant information about the study design, relevant dates, and study location under this subheading. 

c. How did you determine the sample size?

d. I would suggest adding a paragraph about the study’s limitations at the end of the discussion section.  

Author Response

Reviewer 1:

I had the opportunity to review this interesting study evaluating the effects of the complexity of cognitive tasks on gait in a sample of healthy adults. The manuscript is well-written, the study is novel and relevant, and it is methodologically sound. I have only four minor comments to improve the manuscript’s quality. 

1) Please add 1-2 sentences about the study’s methods in the abstract. 

This has now been added –

Line 15-17 “Participants completed both the CBT and WalCT, where they were asked to replicate increasingly complex sequences until they were no longer able to so correctly. IMU sensors were worn on the shins throughout the WalCT to assess changes in gait as task complexity increased.”

2) I would suggest adding a “design” subheading before the “participants” subheading in the methods section and adding the relevant information about the study design, relevant dates, and study location under this subheading. 

This has now been added –

Line 114-117: “2.1. Study Design. This study followed a cross-sectional, within groups design. Data was collected in one of the biomechanics laboratories at Nottingham Trent University between June 2021 – May 2022.”

3) How did you determine the sample size?

This has now been added –

Line 117-119: “Power analysis using G-Power software showed that 36 is sufficient to gather 80% power with a=0.05. To account for attrition, we aimed to recruit ~40 participants for this study.”

4) I would suggest adding a paragraph about the study’s limitations at the end of the discussion section. 

This has been added –

Line 389 – 396: “There are a few notable limitations within this study. Firstly, the WalCT is still a lab-based study and therefore may not relate fully to an individual’s living situation. Additional environmental cues which exist in a standard living environment may assist performance, however this has yet to be studied. Secondly, the population used within this study were healthy adults who typically would not be expected to display any issues with gait or the types of memory being tested. To fully understand this connection, a wider range of participants should be used, e.g. older adults, or those with physical/cognitive impairments.”

Reviewer 2 Report

The manuscript titled “The effect of cognitive task complexity on healthy gait in the Walking Corsi Test” proposes an original investigation on navigational and visual spatial memory as assessed by a single test, namely the Walking Corsi Test (WalCT). Indeed, such test provides the opportunity to assess both types of memory as well as it allows measures of gait to collected. The aim of the study is to investigate whether there is an interplay between cognitive and motor functions as the difficulty of the WalCT trials increase. Forty-three healthy adults were recruited and they were required to complete both the Corsi Block tapping test and the WalCT. Results showed significant differences in several gait parameters between the first three trials and the trial where each participant failed. Authors concluded that the type of memory seemed to have an impact on some gait parameters, and thus that gait is affected by the cognitive task complexity. Authors framed their results in light of previous literature and gave some hints for future research.

I carefully read the manuscript and I think it could be of interest to the readers of Brain Sciences. The manuscript is well-written and properly addresses the interesting issue of the interplay between cognitive functions, especially memory, and motor functions related to walking activities. The paradigm and the main task employed are accurate, as well as the collected measures, and properly pinpoint the constructs to be measured. The introduction section and the aims of the study are clear and detailed. The methodology is very rigorous and accurate, as well as the explanations provided in the discussion section. I also appreciated the general discussion in which Authors provide several links with ageing, implying that further research could address such topics in older adults and in more ecological contexts.

I have few considerations or remarks, which can be found below.

Abstract section

line 15: please provide the full wording for the acronym IMU

Materials and Method section

line 173-180: in those lines you stated that for statistical analysis “the first three successful sequences in each condition and the last failed trial” were considered “to investigate the effect of cognitive task complexity on gait”. Please, can you provide an explanation for using the first three trials and not, for example, only the first one or the first two trials?

line 210: please also add the effect size measure for ANOVA/ANCOVA models, namely partial eta squared.

Results section

line 223: the words “Table 1)” are repeated two times, please remove them. Moreover, since we are dealing with discrete measures, it would be appropriate to report median and IQR (interquartile range) as measures of central location and dispersion. In this view, it would be also appropriate to perform statistics for rank difference instead of ANOVA model. Alternatively, you could report median and IQR in addition to mean and standard deviation.

Author Response

Reviewer 2:

The manuscript titled “The effect of cognitive task complexity on healthy gait in the Walking Corsi Test” proposes an original investigation on navigational and visual spatial memory as assessed by a single test, namely the Walking Corsi Test (WalCT). Indeed, such test provides the opportunity to assess both types of memory as well as it allows measures of gait to collected. The aim of the study is to investigate whether there is an interplay between cognitive and motor functions as the difficulty of the WalCT trials increase. Forty-three healthy adults were recruited and they were required to complete both the Corsi Block tapping test and the WalCT. Results showed significant differences in several gait parameters between the first three trials and the trial where each participant failed. Authors concluded that the type of memory seemed to have an impact on some gait parameters, and thus that gait is affected by the cognitive task complexity. Authors framed their results in light of previous literature and gave some hints for future research.

I carefully read the manuscript and I think it could be of interest to the readers of Brain Sciences. The manuscript is well-written and properly addresses the interesting issue of the interplay between cognitive functions, especially memory, and motor functions related to walking activities. The paradigm and the main task employed are accurate, as well as the collected measures, and properly pinpoint the constructs to be measured. The introduction section and the aims of the study are clear and detailed. The methodology is very rigorous and accurate, as well as the explanations provided in the discussion section. I also appreciated the general discussion in which Authors provide several links with ageing, implying that further research could address such topics in older adults and in more ecological contexts.

I have few considerations or remarks, which can be found below.

1) Abstract section

1a. line 15: please provide the full wording for the acronym IMU

This has been changed

            Line 15: “inertial measurement unit”

2) Materials and Method section

2a. line 173-180: in those lines you stated that for statistical analysis “the first three successful sequences in each condition and the last failed trial” were considered “to investigate the effect of cognitive task complexity on gait”. Please, can you provide an explanation for using the first three trials and not, for example, only the first one or the first two trials?

This has been addressed. Please see lines 186-192: “This approach allowed the differences in gait to be compared between conditions with a low cognitive load initially (the initial successful trials) to initially identify whether there were significant differences in natural gait parameters independent of the task itself. The subsequent comparison between conditions with a low cognitive load (initial successful trials) and high cognitive load (last failed level) allowed the effect of cognitive load on healthy gait to be determined.”

2b. line 210: please also add the effect size measure for ANOVA/ANCOVA models, namely partial eta squared.

The text has been revised and now includes eta-squared values (η2). These have all been highlighted in yellow within the manuscript, both in the relevant tables and text.

We used eta square (η2) as this measures the proportion of the total variance in a dependent variable that is associated with the membership of different groups defined by an independent variable. Partial eta squared is a similar measure in which the effects of other independent variables and interactions are partialled out.

3) Results section

3a. line 223: the words “Table 1)” are repeated two times, please remove them. Moreover, since we are dealing with discrete measures, it would be appropriate to report median and IQR (interquartile range) as measures of central location and dispersion. In this view, it would be also appropriate to perform statistics for rank difference instead of ANOVA model. Alternatively, you could report median and IQR in addition to mean and standard deviation.

We believe that reporting the mean and standard deviation is appropriate for the continuous data presented (the gait variables), however median and IQR have been reported for the discrete measures (the block span) – see table 1